# Drosophila pain sensitization and modulation unveiled by a novel pain model and analgesic drugs

**Wijeong Jang©, Myungsok Oh©, Eun-Hee Cho, Minwoo Baek, Changsoo Kim©** *

School of Biological Sciences and Technology, Chonnam National University, Gwangju, Korea

© These authors contributed equally to this work.
* changgk2001@hanmail.net

**Data Availability Statement:** All relevant data are within the paper and its supporting information files.

## Abstract

In mammals, pain is regulated by the combination of an ascending stimulating and descending inhibitory pain pathway. It remains an intriguing question whether such pain pathways are of ancient origin and conserved in invertebrates. Here we report a new Drosophila pain model and use it to elucidate the pain pathways present in flies. The model employs transgenic flies expressing the human capsaicin receptor TRPV1 in sensory nociceptor neurons, which innervate the whole fly body, including the mouth. Upon capsaicin sipping, the flies abruptly displayed pain-related behaviors such as running away, scurrying around, rubbing vigorously, and pulling at their mouth parts, suggesting that capsaicin stimulated nociceptors in the mouth via activating TRPV1. When reared on capsaicin-containing food, the animals died of starvation, demonstrating the degree of pain experienced. This death rate was reduced by treatment both with NSAIDs and gabapentin, analgesics that inhibit the sensitized ascending pain pathway, and with antidepressants, GABAergic agonists, and morphine, analgesics that strengthen the descending inhibitory pathway. Our results suggest Drosophila to possess intricate pain sensitization and modulation mechanisms similar to mammals, and we propose that this simple, non-invasive feeding assay has utility for high-throughput evaluation and screening of analgesic compounds.

## Introduction

Upon exposure to painful stimuli, both vertebrates and invertebrates commonly exhibit coordinated escape or avoidance behaviors. Noxious substances trigger pain sensors/channels located on specialized sensory neurons (nociceptors) in the peripheral nervous system, which in turn activate ascending nociceptive neural circuits that relay signals from those neurons through spinal sites to pain centers in the brain, where they are interpreted as unpleasant pain [1–4]. Studies mostly in mammals have also identified descending anti-nociceptive pathways that modulate the ascending pain signals at the spinal sites, prior to the signal reaching higher neural centers [5–8].

Tissue damage and nerve injuries can sensitize the ascending neural circuits and inhibit the descending neural circuits, which leads to reduction of pain thresholds (allodynia) and

**Funding:** National Research Foundation (NRF) of Korea (www.nrf.re.kr) to CK (2021R1A2C1010334) and WJ (2020R1I1A1A01074292).

**Competing interests:** The authors have declared that no competing interests exist.

exaggeration of pain response (hyperalgesia) [9–14]. Such sensitization has also been reported in *Drosophila*; for example, a UV-irradiated larval model demonstrated that *Drosophila* nociceptors can be sensitized upon tissue damage, leading to heightened noxious response to both innocuous and noxious stimuli [15]. Recently, a leg amputation model also showed that *Drosophila* develops long-term neuropathic pain states due to abolition of GABAergic modulatory neural circuits from the central nervous system, which is similar to the descending modulatory pain circuits in mammals [16]. However, the molecular and neural mechanisms regarding ascending pain sensitization and especially descending inhibitory neural circuits are still poorly understood in invertebrates.

A variety of analgesic drugs are known to act in distinct ways to reduce sensitized neural pain circuits. Nonsteroidal anti-inflammatory drugs (NSAIDs) and 1-(aminomethyl)cyclohexaneacetic acid (gabapentin) act by reducing afferent nociceptor sensitization and synaptic transmission, respectively [17–20]. Other analgesics such as anti-depressant drugs, GABAergic agonists, and morphine act by enhancing anti-nociceptive signaling and so reducing pain transmission at the central sites of nociceptive pathways [12, 21–24]. Just as the pain circuits in *Drosophila* are yet poorly understood, so is pain medication in *Drosophila* also poorly advanced, in part due to a lack of proper pain models. Here we elicit nociception in *Drosophila* with repeated stimulation of nociceptors and test the responsiveness of this model to pain medications administered in humans, thereby gaining insight into *Drosophila* pain pathways.

## Materials and methods

### Chemicals and preparation of drug-containing foods

Chemicals were purchased from Sigma-Aldrich Co., except for gabapentin and SB366791, which were purchased from Tocris Bioscience. An obstruent drug (active ingredient berberine) and a digestive drug (active ingredient pancreatin) were purchased from a local pharmacy in Korea. Fly food contained 90.6 g dextrose, 68 g dry yeast, 42.8 g corn meal, 6.5 g agar, 0.1% Tegasept, and 4.5 ml propionic acid per liter, which were all dissolved by boiling. The resulting food was then mixed with capsaicin and the various other drugs as appropriate and aliquoted into vials.

### Genetics and fly strains

*Drosophila* stocks were raised on standard fly food medium described above at 25˚C under a 12-h light/dark cycle. All experiments involving drugs were performed with 5-day old adult males, which were tested in groups of 20 per vial. The *md-Gal4* flies were obtained from the Jan Laboratory at the University of California, San Francisco. To produce transgenic flies carrying the *UAS-TRPV1* construct, the coding region of *TRPV1* [25] was amplified by polymerase chain reaction (PCR) using turbo *Pfu* (Stratagene) and cloned into the EcoR1-XbaI sites of the pUAST transformation vector. Red-eyed flies were recovered by microinjection of the *UAS-TRPV1* vector. The presence of *UAS-TRPV1* in those flies was confirmed by PCR using primers corresponding to the *TRPV1* sequence. Primers used for construction of the *UAS-TRPV1* vector were: 5'-GGC GAA TTC ATG AAG AAA TGG AGC-3' (forward), 5'-GGC CTC GAG TCA CTT CTC CCC GGA-3' (reverse). Primers used to confirm the *UAS-TRPV1* transgene were: 5'-ATAGCTCCTA CAACAGCCT-3' (forward), 5'-CACCT GGAACACCAACGT-3' (reverse). All of the selected transgenic lines carried the vector on the third chromosome. Flies bearing a second-chromosomal insertion were obtained from a transgenic line with a third-chromosomal insertion by chromosomal jumping using the *delta 2–3* *P*-element. The fly line with second-chromosomal insertions of both *md-Gal4* and *UAS-TRPV1* was then generated through recombination between *md-Gal4* on the second

chromosome of single-insertion flies and *UAS-TRPV1* on the second chromosome of single-insertion flies. Finally, a strain of transgenic flies that carried *w*; *md-Gal4, UAS-TRPV1/CyO*; *UAS-TRPV1/UAS-TRPV1* was obtained and used for most of the drug experiments.

### Drug feeding experiments

Drugs were dissolved in DMSO and added to fly food to yield a final concentration of 1% DMSO (SB366791, amitriptyline, trazodone, benzodiazepine, carbamazepine, aspirin, acetaminophen, and tolfenamic acid), 2.5% DMSO (gabapentin), 5% DMSO (ibuprofen). Diclofenac was dissolved in 100% ethanol and added to fly food to yield a final concentration of 10% ethanol. Fly food containing 1, 2.5, 5% DMSO, or 10% ethanol was used as a control for the corresponding drug-containing food solutions. Five-day-old male transgenic flies, 20 flies in each vial, were used for the analgesic drug tests. Flies were first fed drug-containing food for 24 h, then transferred to a food source containing 5 mM capsaicin and varying amounts of the drug. For the morphine tolerance test, flies were fed 660 μM morphine-containing food for 0, 1, 3, or 6 days prior to transferring them to a food source containing 5 mM capsaicin and 660 μM morphine. In all assays, the flies were examined each day for viability.

### Data analyses

All data except Fig 1C were analyzed by analysis of variance (ANOVA) using the SPSS statistical software. One-way ANOVA with the Tukey-Kramer method for multiple comparisons was used when warranted. Paired Student's *t*-test was used for Fig 1C.

## Results

### Pain model with repeat nociceptor stimulation

In mammals, the heat and capsaicin receptor TRPV1 mediates heat- and capsaicin-induced pain and also inflammatory and chronic pain [26–32]. In *Drosophila*, heat elicits thermal nociception, but capsaicin does not elicit nociception [33, 34]. To stimulate nociceptors and thereby elicit nociception in *Drosophila* upon capsaicin sipping, we expressed the human capsaicin receptor TRPV1 in nociceptors via the *Gal4/UAS* binary expression system [35]. We generated transgenic flies bearing one copy of *md-Gal4*, in which *Gal4* is expressed in nociceptive multidendritic (md) neurons [36–38], along with one, two, or three copies of the *UAS-TRPV1* construct. All experiments (except for Figs 2D and 3B) were performed with transgenic flies bearing one copy of *md-Gal4* and three copies of *UAS-TRPV1* (henceforth referred to as *md-TRPV1(3)*, in which (3) indicates copy number) due to such flies exhibiting the strongest nociceptive response against capsaicin.

To examine whether capsaicin elicited nociception in *md-TRPV1(3)* larvae, we touched abdominal segment five of the larvae with a brush soaked with capsaicin (20 mM). Wild-type larvae did not respond to the capsaicin brush; in contrast, brush-touched *md-TRPV1(3)* larvae exhibited robust rolling (S1 Video), a readout of larval nociception, suggesting that capsaicin elicited nociception in the transgenic larvae but not wild-type larvae. Additionally, this nociception was dependent on capsaicin concentration (S1A and S1B Fig), indicating that the elicited nociception occurred through activation of the capsaicin receptor TRPV1 in nociceptors of the skin.

The next question was whether capsaicin elicits nociception in adult flies. Wild-type and *md-TRPV1(3)* flies were starved for 18 hours on a water-soaked glass filter and then offered food containing capsaicin (5 mM). Hungry control flies (*md-Gal4* or *UAS-TRPV1(3)*) continued eating the capsaicin-containing food without exhibiting nocifensive behaviors (S2 Fig). In

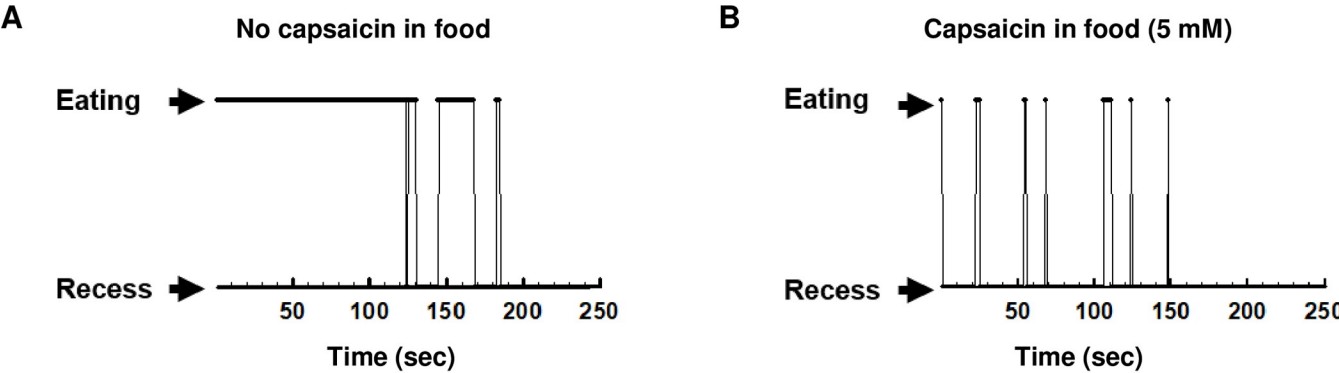

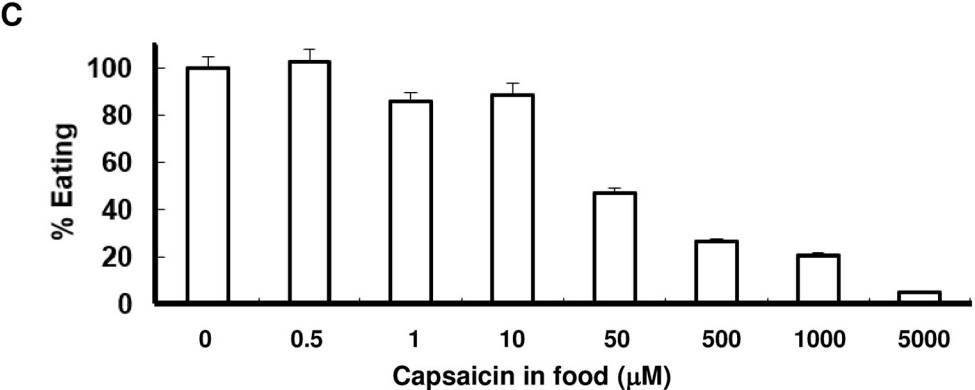

**Fig 1. *md-TRPV1(3)* flies are aversive to capsaicin ingestion.** (A, B) Hungry (18 hours starved on water-soaked filters) *md-TRPV1(3)* flies were offered either normal food (A) or capsaicin-containing food (B). Those provided normal food continuously sipped for longer than three minutes before idling, whereas flies given 5 mM capsaicin-containing food exhibited repeated brief sipping (~ one second) and longer recess intervals. (C) The sipping period of hungry *md-TRPV1(3)* flies on capsaicin food is dependent on capsaicin concentration. Values are the average of eating intervals measured for 2 minutes beginning immediately after flies were transferred to food containing varying amounts of capsaicin (0 to 5 mM). Bars represent mean ± SD. n = 10 for each capsaicin concentration. All flies were 5-day-old males. *md-TRPV1(3)* denotes one copy of *md-Gal4* and 3 copies of *UAS-TRPV1*.

contrast, hungry *md-TRPV1(3)* flies, which continued eating the normal food lacking capsaicin (Fig 1A), sipped the capsaicin-containing food only briefly (~ one second), then ran away, scurried around, and vigorously rubbed and pulled on their mouthparts with their front legs (S2 Video), which looked like an expression of pain. The transgenic flies then returned to their food and sipped again for a brief period (~ one second), followed again by running away; they repeated these unusual behaviors several times (Fig 1B and S3 Fig). Moreover, the sipping periods decreased as capsaicin concentration increased (Fig 1C). Taken together, these behaviors suggest that capsaicin repeatedly stimulated nociceptors abundant in the mouth (S1C Fig) and elicited nociception in the *md-TRPV1(3)* flies, but not in control (*md-Gal4 or UAS-TRPV1(3)*) flies.

Due to this nociception, *md-TRPV1(3)* flies were rarely detectable on capsaicin-containing food sites (S4 Fig), which suggests that *md-TRPV1(3)* flies ultimately starved to death. When provided 5 mM capsaicin-containing food at 29°C, *md-TRPV1(3)* flies exhibited short viability with empty abdomens and weight loss (S5 Fig). The observed viability was similar to that of *md-TRPV1(3)* flies fed only water (S5A Fig), supporting the effect as being due to starvation. However, upon shifting from capsaicin food to normal food, 90% of *md-TRPV1(3)* flies

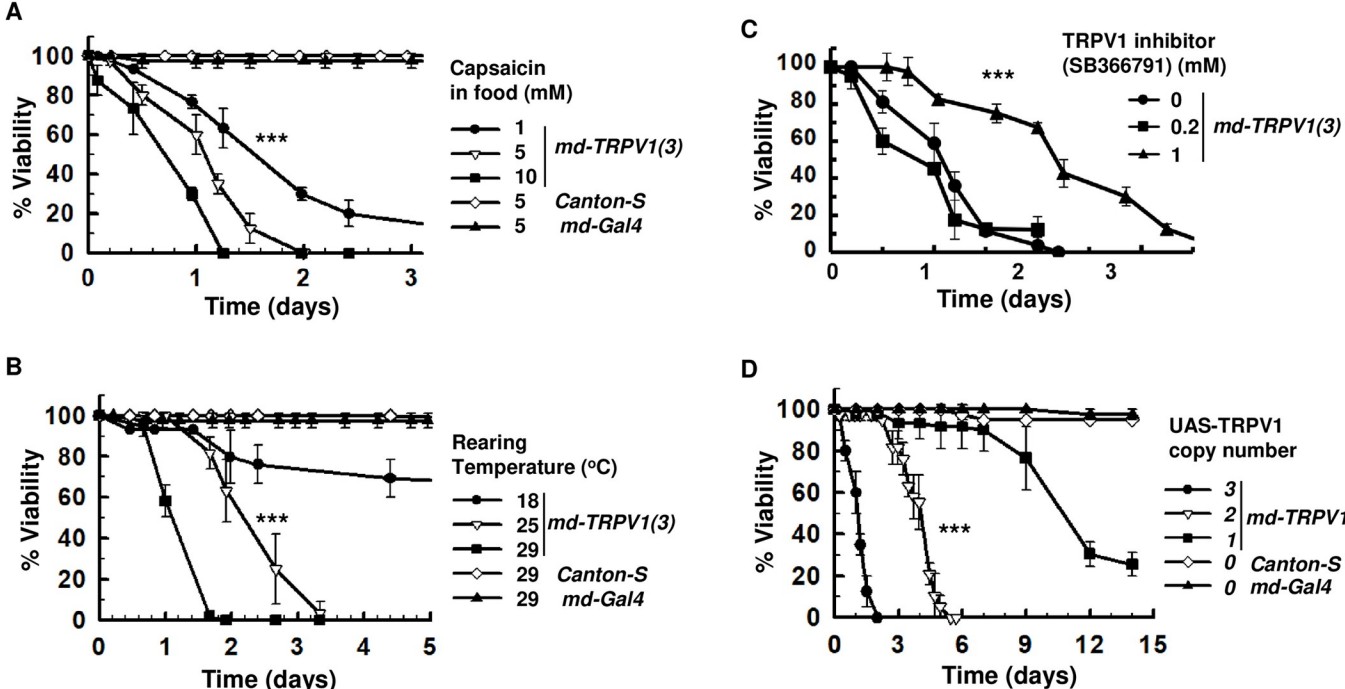

**Fig 2. Effect of capsaicin concentration, rearing temperature, a TRPV1 inhibitor, and *UAS-TRPV1* copy number on the viability of *md-TRPV1(3)* flies grown on capsaicin-containing food.** (A) Viability of *md-TRPV1(3)* flies grown at 29°C on food containing capsaicin at indicated concentrations. Control (*Canton-S* or *md-Gal4*) flies were grown at 29°C on food containing 5 mM capsaicin. Dots and vertical lines denote means and standard deviations, respectively. n = 60 (20 flies per vial) for each capsaicin concentration; one-way ANOVA with the Tukey-Kramer method for multiple comparisons, *** $P < 0.001$ for 1 mM *vs* 10 mM. (B) Viability of *md-TRPV1(3)* flies grown on 5 mM capsaicin-containing food at the temperatures indicated. Control (*Canton-S* or *md-Gal4*) flies were grown at 29°C on food containing 5 mM capsaicin. Dots and vertical lines denote means and standard deviations, respectively. n = 60 for each temperature; one-way ANOVA with the Tukey-Kramer method for multiple comparisons, *** $P < 0.001$ for 29°C *vs* 25°C. (C) Viability of *md-TRPV1 (3)* on food containing capsaicin (5 mM) and SB366791 at the concentrations indicated. Dots and vertical lines denote means and standard deviations, respectively. n = 60 for each curve; one-way ANOVA with the Tukey-Kramer method for multiple comparisons, *** $P < 0.001$ for no SB366791 *vs* 1 mM SB266791. (D) Viability of *md-TRPV1* flies grown on capsaicin (5 mM)-containing food at 29°C. The number denotes the number of *UAS-TRPV1* copies. Dots and vertical lines denote means and standard deviations, respectively; one-way ANOVA with the Tukey-Kramer method for multiple comparisons, *** $P < 0.001$ for two *vs* three copies of *UAS-TRPV1*. Control (*Canton-S* or *md-Gal4*) flies were grown at 29°C on food containing 5 mM capsaicin. *md-TRPV1(3)* denotes one copy of *md-Gal4* and 3 copies of *UAS-TRPV1*. All flies were 5- day-old males.

remained viable (S6 Fig). Taken together, these findings suggest that *md-TRPV1(3)* flies avoid eating capsaicin-containing food and ultimately die of starvation.

To examine whether the viability of *md-TRPV1(3)* flies is dependent on capsaicin concentration, we varied the amount of capsaicin used in their food. Decrease of capsaicin from 10 mM to 1 mM led to two-fold increased viability, with the half-viability period extending from 16 hr. to 38 hr (Fig 2A). Additionally, we examined the effect of rearing temperature on viability of capsaicin-fed *md-TRPV1(3)* flies. Decreasing the rearing temperature from 29°C to 25°C led to two-fold increased viability, with respective half-viability periods of 28 hr and 52 hr (Fig 2B). Further decrease of rearing temperature to 18°C caused the viability to increase profoundly (Fig 2B). Finally, we tested whether inhibition of TRPV1 affects the viability of these transgenic flies. SB366791 is a TRPV1 inhibitor demonstrated to work well when administered orally in rat models of pain [39]; accordingly, we tested the addition of different concentrations of inhibitor to food also containing 5 mM capsaicin. Addition of 0.2 mM SB366791 was not effective, but 1 mM increased viability by two-fold, from a half-viability of 24 hr to 48 hr (Fig 2C); meanwhile, 5 mM was toxic to the flies (S7 Fig). We further explored alternatively reducing TRPV1 activities in md neurons by reducing the copy number of *UAS-TRPV1*.

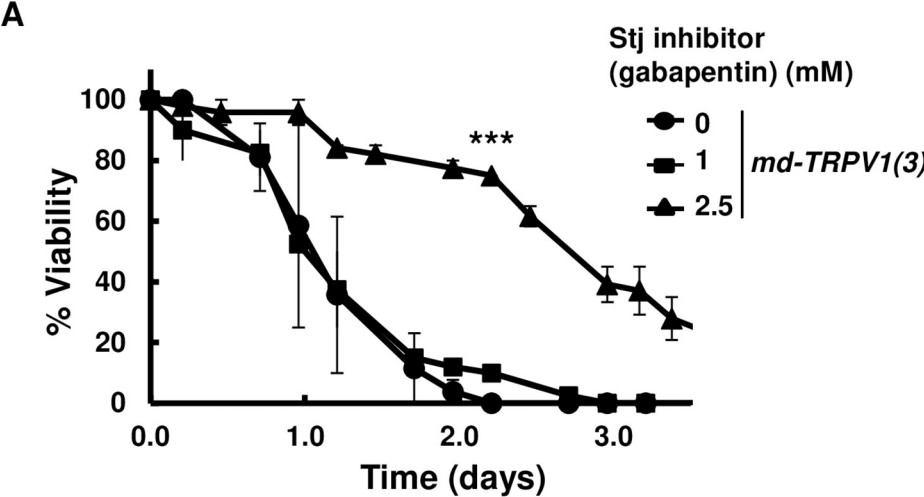

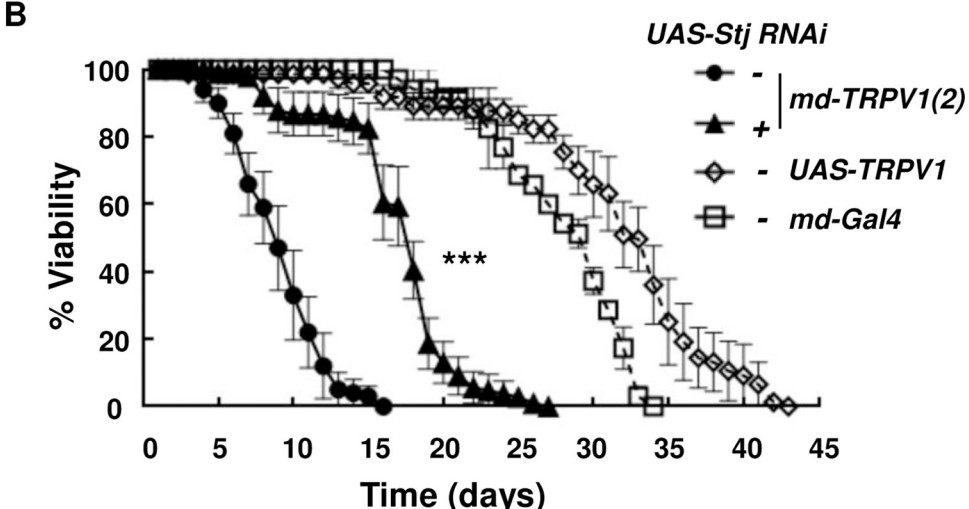

**Fig 3. Effect of gabapentin and its target, *sti*, the $\alpha_2\delta$-subunit of voltage-gated calcium channels, on the viability of *md-TRPV1* flies grown on capsaicin-containing food at 29°C.** (A) Viability of *md-TRPV1(3)* flies reared on food that contained capsaicin (5 mM) and gabapentin at the concentrations indicated. Dots and vertical lines denote means and standard deviations, respectively. n = 60 for each curve; one-way ANOVA with the Tukey-Kramer method for multiple comparisons, *** $P < 0.001$ for no gabapentin *vs* 2.5 mM gabapentin. *md-TRPV1(3)* denotes one copy of *md-Gal4* and 3 copies of *UAS-TRPV1*. (B) Viability of *md-TRPV1(2)* flies bearing one copy of *md-Gal4* and two copies of *UAS-TRPV1* with or without *UAS-stj* RNAi grown on food with capsaicin (5 mM) at 29°C. Other flies (*UAS-TRPV1* and *md-Gal4*) were used as controls. Dots and vertical lines denote means and standard deviations, respectively. n = 60 for each curve; one-way ANOVA with the Tukey-Kramer method for multiple comparisons, *** $P < 0.001$ for *md-TRPV1(2) vs md-TRPV1(2), UAS-stj-RNAi*, in which (2) indicates two copies of *UAS-TRPV1*. All flies were 5-day-old males.

Reducing copy number from three to two increased viability four-fold; reducing further to one copy increased viability eleven-fold (Fig 2D). These confirm the notion that capsaicin-associated viability in transgenic *md-TRPV1* flies is dependent on the magnitude of TRPV1 activation and thus the degree of nociceptor activation and total nociception.

## Efficacy of analgesics presumed to inhibit the ascending pain pathway

Gabapentin, an analgesic for chronic neuropathic pain, acts on the $\alpha_2\delta$-subunit of voltage-gated calcium channels (VGCCs) in mammals [40–44]. Gabapentin was recently shown to be effective at reducing leg amputation neuropathic pain in *Drosophila* by targeting *straight-jacket* (*stj*), which encodes the $\alpha_2\delta$ subunit of *Drosophila* VGCCs [45]. Accordingly, we tested the effect of gabapentin and its target, *stj*, in our nociception model. Adding 1 mM gabapentin to capsaicin food was not effective, but 2.5 mM gabapentin, which did not affect viability of control flies (*md-Gal4*) on normal food lacking capsaicin (S8 Fig), increased viability 3-fold, extending the half-viability period of one day to 2.7 days (Fig 3A). This suggests that gabapentin is effective in reducing the elicited nociception in our model. A higher concentration (5 mM) of gabapentin was toxic to the flies (S7 Fig). *Stj* is expressed in nociceptors, wherein it has been demonstrated to be required for both thermal and neuropathic nociception [45, 46]. Knock-down of *stj* in md neurons via expression of *stj* RNAi doubled the viability of *md-TRPV1(2)* flies (that is, flies with two copies of *UAS-TRPV1*) from a half-viability of 8 days to 17 days (Fig 3B). Taken together, these findings suggest that activity of *stj* is required for nociception as represented by the viability of *md-TRPV1* flies fed capsaicin-containing food.

Non-steroidal anti-inflammatory drugs (NSAIDs) act as analgesics for inflammatory pain through inhibiting synthesis of prostaglandin, an enhancer of nociceptor sensitization [47–51]. Adding NSAIDs to capsaicin food increased the viability of *md-TRPV1(3)* flies, suggesting that NSAIDs are effective in reducing the elicited pain in this model. Of those tested, diclofenac and ibuprofen were the most effective, respectively, increasing viability 5- and 6-fold (Fig 4) without affecting the viability of control flies on normal food lacking capsaicin (S8 Fig). Higher concentrations of NSAIDs were toxic (S7 Fig).

## Efficacy of analgesics presumed to enhance the modulatory pain pathway

Neuropathic pain medication relives pain through strengthening descending inhibitory circuits [12, 52–54]. We tested diverse neuropathic pain medications in our pain model. Adding antidepressants to capsaicin food increased the viability of *md-TRPV1(3)* flies. Antidepressants increased viability, by 3-fold for amitriptyline and 6-fold for trazodone (Fig 5A and 5B); neither affected the viability of control flies on normal food lacking capsaicin (S8 Fig). The gamma-aminobutyric acid (GABA) receptor agonists benzodiazepine increased viability by 3-fold (Fig 5C), without affecting viabilities of control (*md-Gal4*) flies (S8 Fig). Morphine also increased viability by 3-fold (Fig 6A). Notably, prolonged or repeated use of morphine decreases its efficacy [22, 55, 56]. To examine if prolonged use of morphine decreases its efficacy in our pain assay, a morphine pretreatment was applied. When flies were pre-fed morphine (660 µM)-containing food for just one day, no effect of morphine (660 µM) on viability was observed; in contrast, flies that were pre-fed morphine (660 µM)-containing food every day for six days showed significantly reduced analgesic effects, suggesting that morphine pretreatment decreased its efficacy (Fig 6B). In contrast to analgesic drugs, fly viability was not affected by non-analgesic drugs, including isoproterenol (used to treat asthma), hydrochorothiazide (an antihypertensive diuretic), and D-mannitol (an osmotic diuretic) (S9 Fig).

## Discussion

Here we elicited nociception in *Drosophila* via repeated stimulation of nociceptors and tested the efficacy of diverse analgesic medications that are routinely administered in inflammatory and neuropathic human pain. We show that the elicited pain is reduced by NSAIDs and

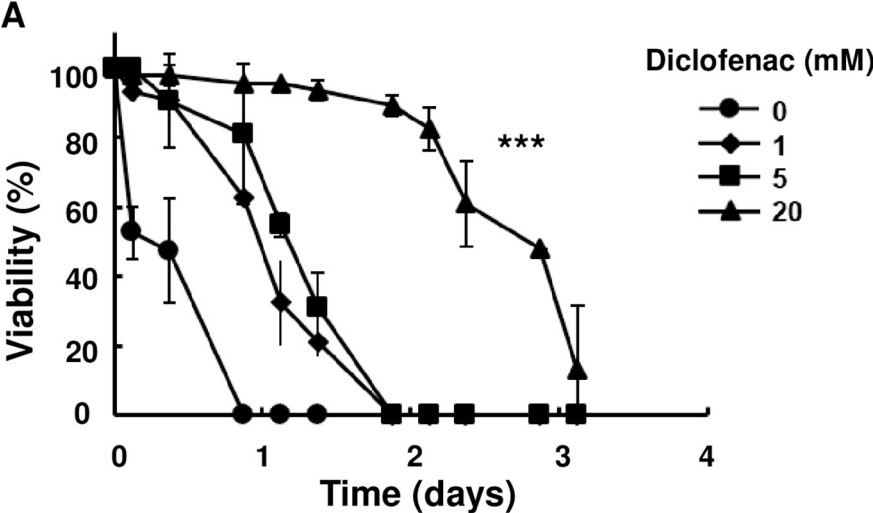

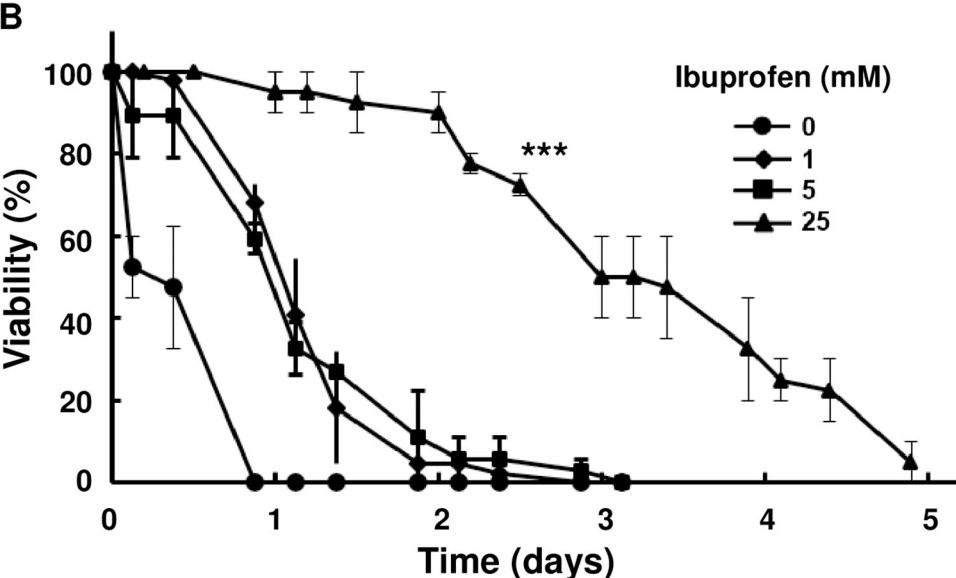

**Fig 4. Effect of NSAIDs (diclofenac and ibuprofen) on the viability of *md-TRPV1(3)* flies grown on capsaicin (5 mM)-containing food at 29˚C.** Drugs were added to food at the concentrations indicated in the figure. Dots and vertical lines denote means and standard deviations, respectively. n = 60 for each curve; one-way ANOVA with the Tukey-Kramer method for multiple comparisons, *** $P < 0.001$ for no diclofenac *vs* 20 mM diclofenac and for no ibuprofen *vs* 25 mM ibuprofen. All flies were 5-day-old males. *md-TRPV1(3)* denotes one copy of *md-Gal4* and 3 copies of *UAS-TRPV1*.

gabapentin, which are presumed to reduce nociceptor sensitization and synaptic transmission, respectively, and also by GABAergic agonists, antidepressants, and morphine, which are presumed to enhance the modulatory descending pain pathway. Therefore, our findings suggest that the molecular mechanisms and neural networks of nociception sensitization and modulation are highly conserved across both vertebrates and invertebrates and that pain pathways in *Drosophila* are as complex as those of the mammalian pain system.

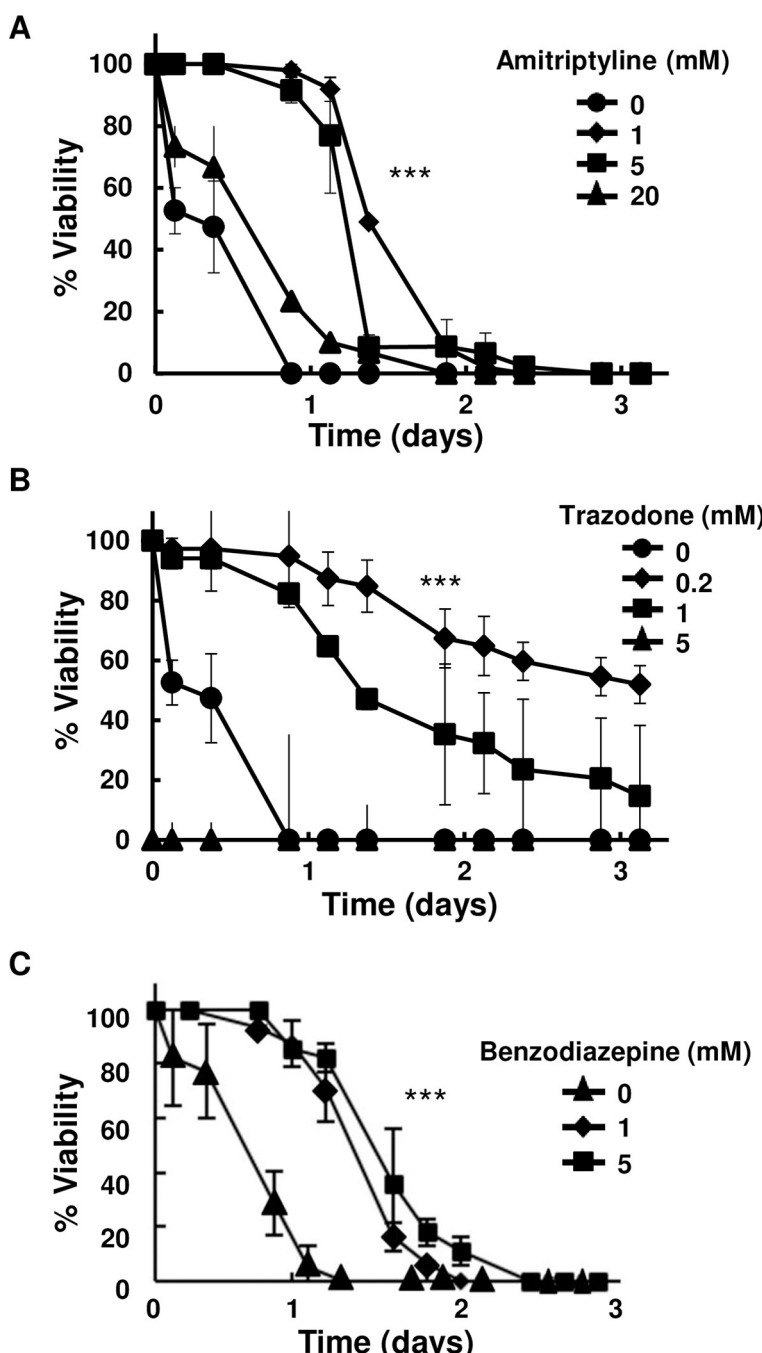

**Fig 5. Effect of antidepressants (amitriptyline and trazodone) and GABAergic receptor agonists (benzodiazepine) on the viability of *md-TRPV1(3)* flies grown on capsaicin (5 mM)-containing food at 29°C.** Drugs were added to food at the concentrations indicated. Dots and vertical lines denote means and standard deviations, respectively. n = 60 for each curve; one-way ANOVA with the Tukey-Kramer method for multiple comparisons, *** $P < 0.001$ for no drug *vs* 1 mM amitriptyline, 0.2 mM trazodone, 5 mM benzodiazepine. All flies were 5-day-old males. *md-TRPV1(3)* denotes one copy of *md-Gal4* and 3 copies of *UAS-TRPV1*.

A

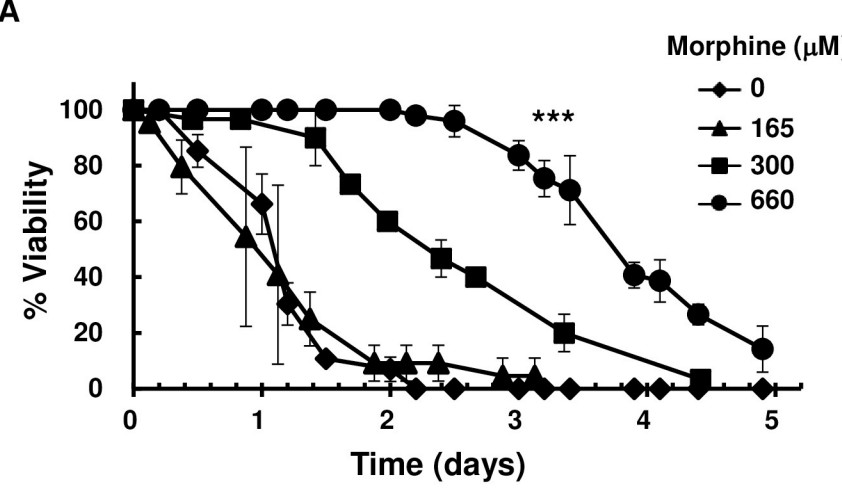

B

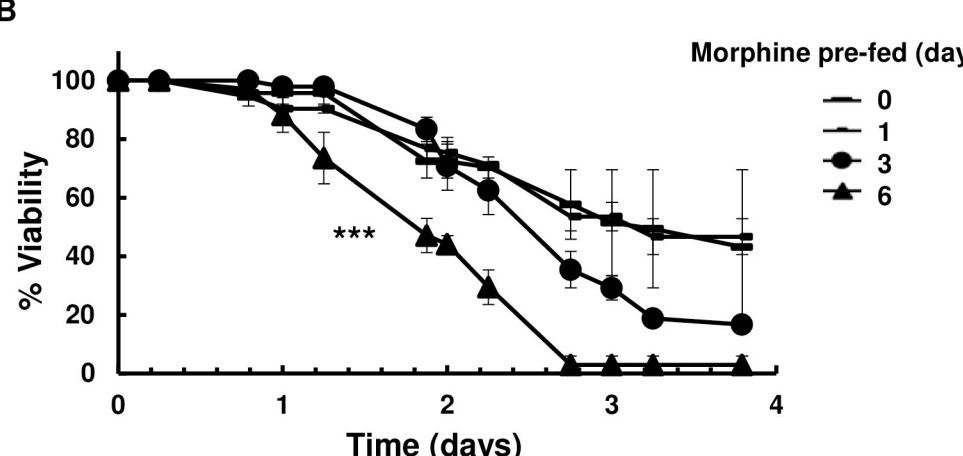

**Fig 6. Effect of morphine on the viability of *md-TRPV1(3)* flies grown on capsaicin (5 mM)-containing food at 29˚C.** (A) Viability of *md-TRPV1(3)* flies grown on food with morphine added at the concentrations indicated. Dots and vertical lines denote means and standard deviations, respectively. n = 60; one-way ANOVA with the Tukey-Kramer method for multiple comparisons, *** $P < 0.001$ for no morphine *vs* 660 μM morphine. (B) Viability of *md-TRPV1(3)* flies that were pre-fed with morphine (660 μM)-containing food for one, three, or six days and then transferred to a food that contained 5 mM capsaicin and 660 μM morphine. Dots and vertical lines denote means and standard deviations, respectively. n = 60 for each curve; one-way ANOVA with the Tukey-Kramer method for multiple comparisons, *** $P < 0.001$ for no pre-feeding *vs* six days pre-fed. All flies were 5-day-old males. *md-TRPV1(3)* denotes one copy of *md-Gal4* and 3 copies of *UAS-TRPV1*.

### Pain model of repeat nociceptor stimulation

The pain model presented here employs transgenic flies that express the human capsaicin receptor TRPV1 in nociceptors. This invites the question as to whether capsaicin stimulated the nociceptors and elicited nociception in the animals. Our results suggested that capsaicin did indeed stimulate the nociceptors and thereby elicit nociception via TRPV1 activation in both transgenic larvae and flies. Firstly, capsaicin touch on the skin of *md-TRPV1(3)* larvae evoked rolling, which is a readout of larval nociception. Secondly, capsaicin touch on the tongue of *md-TRPV1(3)* flies generated abrupt escape from the food source and pain-expressing behaviors like rubbing and pulling at their mouth parts. Critically, larval and adult nocifensive responses alike were associated with capsaicin concentration and *UAS-TRPV1* copy number,

both of which are related to the magnitude of TRPV1 activation: the greater the capsaicin and *UAS-TRPV1* copy number, the stronger the nociceptive response exhibited by *md-TRPV1* larvae and flies. These findings support that capsaicin activated the nociceptors and thereby elicited nociception via stimulation of TRPV1 in these transgenic larvae and flies.

Hungry (starved) *md-TRPV1(3)* flies sipped on capsaicin (5 mM)-containing food for one second and then abruptly escaped from the food site; this behavior was not seen in control flies (*md-Gal4* or *UAS-TRPV1(3)*). Being still hungry, they returned to sip briefly (one second) and escape again. These sipping actions would stimulate nociceptors repeatedly, which produces nociceptor sensitization, leading to hyperalgesia and allodynia [10, 57–60]. Of note, the flies were rarely detectable on the food site, suggesting that the short viability of *md-TRPV1(3)* flies on capsaicin food is due to lack of food intake. Remarkably, fly viability was dependent on parameters of TRPV1 activation, namely the capsaicin concentration in the food and *UAS-TRPV1* copy number, suggesting that the observed viability represents the magnitude of nociceptor stimulation via TRPV1 activation and, therefore, the pain experience of the flies upon capsaicin sipping. This notion is further supported by the increase of viability observed when a TRPV1 inhibitor was added to the capsaicin food. Consequently, this feeding assay could be used to identify analgesics, which are added to the capsaicin food, based on increased viability.

### Efficacy of analgesics presumed to reduce the ascending pain pathway

With this feeding assay, we found that gabapentin, an analgesic administered for neuropathic pain [40, 43, 61, 62], also reduced the elicited pain in the flies. Gabapentin works through inhibiting the $\alpha_2\delta$-subunit of VGCCs, which is essential to synaptic transmission [40, 63–65]. Consistent with this function, we also found that RNAi-mediated knockdown of *stj*, the *Drosophila* homolog of the mammalian $\alpha_2\delta$-subunit of VGCCs, in md neurons also reduced the elicited pain. These results are in accord with a recent finding in both *Drosophila* and mice that *stj* is required in nociceptors, wherein it mediates thermal nociception and neuropathic pain [44–46, 63]. It is noteworthy that gabapentin was recently shown to be effective in reducing neuropathic pain in a *Drosophila* leg-amputation pain model, and moreover to produce this effect through inhibiting *stj* [45]. Taken together, it is clear that reducing nociceptor transmission either with a drug (gabapentin) or by knockdown of its target gene (*stj*) reduced the elicited pain.

In addition, we found that non-steroidal anti-inflammatory drugs (NSAIDs), analgesics administered in inflammatory pain, reduced the elicited pain in this pain model. NSAIDs are effective in reducing nociceptor sensitization and firing via their inhibition of cyclooxygenase (COX) [66]. In inflammatory pain, COX produces prostaglandins (PGs) that sensitize nociceptors, leading to hypersensitivity to both innocuous and noxious stimuli [66, 67]. Our findings suggest that the elicited pain in *md-TRPV1(3)* flies involves PG-mediated nociceptor sensitization. Consistent with this hypothesis, cyclooxygenase-like activity has been reported in fly and in insect tissue extracts [68–71], and NSAIDs documented to block PG synthetic activity in insect tissue extracts [72, 73]. Our findings along with other observations suggest that *Drosophila* might possess pain sensitization mechanisms based on COX and prostaglandins, similar to those in mammals.

### Efficacy of analgesics presumed to enhance the modulatory pain pathway

Gamma-aminobutyric acid (GABA) receptor agonists are effective in reducing pain, particularly neuropathic pain, through enhancing descending inhibitory GABAergic neural circuits [21, 74–76]. We found that GABA receptor agonists, which enhance the action of GABAergic neural circuits, reduced the elicited pain in this assay. Notably, a *Drosophila* leg amputation model recently showed GABAergic inhibitory neural circuits to subsequently be removed from the fly central nervous system, which results in a chronic pain state [16]. Taken together,

our findings and those of others support the notion that GABAergic modulatory nociceptive neural circuits constitute descending inhibitory nociceptive neural circuits in *Drosophila*.

Antidepressants likewise alleviate pain by enhancing the descending inhibitory pathway through serotonin (5-hydroxytryptamine, 5-HT) circuits at the central sites of nociceptive pathways [5, 12, 74, 77–80]. Here we show that antidepressants relive the elicited pain in *md-TRPV1 (3)* flies, suggesting that *Drosophila* might have anti-nociceptive descending neural circuits similar to those in mammals that act to modulate pain. This conjecture is further strengthened by the observed relief of nociception by morphine, which is presumed to act through morphinergic neuronal circuits that constitute an integral part of the anti-nociceptive neural pathway [23, 24, 81]. Therefore, flies might possess anti-nociceptive signaling pathways involving both serotonergic and morphinergic neural circuits, similar to the system in mammals.

### *Drosophila* molecular targets of analgesic drugs

A multitude of currently-used analgesic drugs are effective in reducing pain in our pain model; however, the molecular targets of these analgesics in *Drosophila* are poorly understood. Cyclooxygenases (COXs), the established molecular targets of NSAIDs, are not well documented in flies. Recently, rigorous bioinformatics approaches involving iterative sequence searches with tertiary structural modeling have identified three putative fly COXs (CG4009, CG6969, Pxt) [82]. It is an intriguing question as to whether these putative COXs mediate inflammatory pain sensitization in *Drosophila*. Also, intriguing is the question of whether the *Drosophila* GABA receptor and serotonin transporter (dSERT), the respective targets of GABA agonists and antidepressants, respectively, are involved in *Drosophila* pain. Additionally, morphine receptors have not been identified in *Drosophila*, but a few reports suggest their presence. Morphine has been shown to mediate pain reduction (in our study). Endo-morphine-like molecules are known to be released in the neurons of some invertebrates [83, 84] and the *Drosophila* head membrane binds morphine with high affinity [85]. Still, it is unknown what morphine-like-receptor might exist and mediate pain in *Drosophila*.

## Conclusions

The pain assay described here is based on elicited pain with repeated stimulation of nociceptors by capsaicin sipping in transgenic flies that express the human heat and capsaicin receptor (TRPV1) in nociceptors. The magnitude of the elicited pain is represented as the viability of flies on capsaicin-containing food. The system is 'sensitized,' as a slight perturbation of the noxious stimulus profoundly affects the pain experienced by the transgenic flies and thus their viability; for example, an additional copy of *UAS-TRPV1* yielded a seven-day difference. As demonstrated by the drug efficacy assays performed in this work, our system can be used to measure the effects of analgesic drugs as long as the agent has a slight influence on the nociception experienced by the transgenic flies. Moreover, this simple feeding assay allowed us to show that both analgesics that reduce afferent sensitized pain and those that strengthen descending modulatory circuits are effective in reducing *Drosophila* pain, suggesting that *Drosophila* nociception features intricate pain sensation, sensitization, and modulatory neural circuits comparable to those in mammals. We propose that this feeding assay, which is sensitive and non-invasive, can be utilized for rapid evaluation and screening of candidate compounds (or lead compounds) to identify those having analgesic effects *in vivo*.

## Supporting information

**S1 Video. Capsaicin induces rolling in *md-TRPV1(3)* larvae.** A brush soaked with 20 mM capsaicin solution was touched to the skin of *md-TRPV1(3)* larvae and the response recorded

with a video camera. *md-TRPV1(3)* denotes one copy of *md-Gal4* and 3 copies of *UAS-TRPV1*.
(AVI)

**S2 Video. Behavior of *md-TRPV1(3)* flies after sipping on capsaicin-containing food.** After being starved for 18 hours, a *md-TRPV1(3)* fly was transferred to 5 mM capsaicin-containing food and the response recorded with a video camera. *md-TRPV1(3)* denotes one copy of *md-Gal4* and 3 copies of *UAS-TRPV1*.
(AVI)

**S1 Fig. Nociceptive rolling response of *md-TRPV1(3)* larvae upon exposure to capsaicin solution and *Gal4* expression in the mouthpart of *md-Gal4* fly.** (A, B) Aversive rolling response of *md-TRPV1(3)* larvae. A capsaicin-soaked brush contacts the abdominal segment five; rolling within 10 seconds was counted as response. n = 30 for each capsaicin concentration. Error bars indicate ±SEM of more than three independent experiments. $w^{1118}$ larvae were used as a control. Paired t-test, *** $P < 0.001$ for no capsaicin *vs* 20 mM capsaicin. (C) A confocal image of the mouthpart of a *md-Gal>UAS-mCD8GFP* fly stained with anti-GFP antibodies to indicate presence of *md* neurons. *md-TRPV1(3)* denotes one copy of *md-Gal4* and 3 copies of *UAS-TRPV1*.
(PPTX)

**S2 Fig. Control flies are not aversive to capsaicin ingestion.** Hungry (18 hours starved on water-soaked filters) control (*md-Gal4* and *UAS-TRPV1(3)*) flies were offered capsaicin (5 mM)-containing food. The flies continuously sipped for longer than five minutes. Representative behaviors are shown. Five flies exhibited similar behaviors. Five-day-old males were used.
(PPTX)

**S3 Fig. *md-TRPV1(3)* flies are aversive to capsaicin ingestion.** Hungry (18 hours starved on water-soaked filters) *md-TRPV1(3)* flies were offered capsaicin (5 mM)-containing food, on which they exhibited repeated brief sipping (~ one second) and longer recess intervals. The sipping behaviors of three flies over 10 min are shown. *md-TRPV1(3)* denotes one copy of *md-Gal4* and three copies of *UAS-TRPV1*. Five-day-old males were used.
(PPTX)

**S4 Fig. *md-TRPV1(3)* flies are aversive to capsaicin-containing food.** Hungry (18 hours starved on water-soaked filters) *md-TRPV1(3)* flies were offered either capsaicin-containing food (5 mM; left vials) or normal food lacking capsaicin (right vials). Flies given capsaicin-containing food were rarely detectable on the food site, while those provided normal food were frequently detectable on the food site. *md-TRPV1(3)* denotes one copy of *md-Gal4* and three copies of *UAS-TRPV1*. Five-day-old males were used.
(PPTX)

**S5 Fig. Aversive behavior of *md-TRPV1(3)* flies grown on capsaicin (5 mM)-containing food.** (A) Viabilities of *md-TRPV1(3)* reared on capsaicin-containing food or water-soaked filters at 29˚C. *md-Gal4* flies were used as control. n = 40 for each experiment. (B) Empty abdomen of a *md-TRPV1(3)* fly that was provided capsaicin (5 mM) food and full abdomen of one provided normal food for 36 hours at 25˚C. n = 10. (C) Weight loss of *md-TRPV1(3)* flies on capsaicin (5 mM) food. Weight was measured after transfer to normal food or capsaicin (5 mM) food for 21 hours and 28 hours, respectively, at 25˚C. n = 40 for each point. Five-day-old males were used. *md-TRPV1(3)* denotes one copy of *md-Gal4* and 3 copies of *UAS-TRPV1*.
(PPTX)

**S6 Fig. Requirement of continuous rearing on capsaicin-containing food for the death of**
*md-TRPV1(3)* **flies.** The viability of *md-TRPV1(3)* flies reared on capsaicin-containing food
(circles) *vs* that of *md-TRPV1(3)* flies reared for two days on capsaicin-containing food and
then transferred to normal food lacking capsaicin (squares). The day of transfer is indicated by
a red arrow. Dots and vertical lines denote means and standard deviations, respectively. n = 60
(20 flies per vial) for each curve. Five-day-old males were used. *md-TRPV1(3)* denotes one
copy of *md-Gal4* and three copies of *UAS-TRPV1*.
(PPTX)

**S7 Fig. Dependency of** *md-TRPV1(3)* **fly viability on drugs in a concentration-dependent**
**manner.** The viability of *md-TRPV1(3)* flies reared on capsaicin (5 mM)-containing food sup-
plemented with varying amounts of drugs marked in the curve is dependent on the concentra-
tion of the supplemented drugs. *md-TRPV1(3)* denotes one copy of *md-Gal4* and three copies
of *UAS-TRPV1*. Five-day-old males were used.
(PPTX)

**S8 Fig. Effect of analgesic drugs on the viability of the control flies.** The viability of control
(*md-Gal4*) flies fed normal food supplemented with analgesic drugs (at the concentration most
effective in pain reduction) reared at 29˚C is shown. Five-day-old males were used. n = 60 (20 flies
per vial). The dots and vertical lines denote the means and standard deviations, respectively.
(PPTX)

**S9 Fig. Effect of non-analgesic drugs on** *md-TRPV1(3)* **fly viability.** Drugs were added to
food at the concentration indicated. Viability of *md-TRPV1(3)* flies on capsaicin (5 mM) con-
taining food supplemented with non-analgesic drugs at 29˚C. Dots and vertical lines denote
means and standard deviations, respectively. n = 60 for each curve. Five-day-old males were
used. *md-TRPV1(3)* denotes one copy of *md-Gal4* and 3 copies of *UAS-TRPV1*.
(PPTX)

## Acknowledgments

We are grateful to Y. Jan (UCSF) for the *md-Gal4* line.

## Author Contributions

**Conceptualization:** Wijeong Jang, Myungsok Oh, Changsoo Kim.

**Data curation:** Wijeong Jang, Myungsok Oh, Eun-Hee Cho, Minwoo Baek.

**Formal analysis:** Changsoo Kim.

**Funding acquisition:** Wijeong Jang, Changsoo Kim.

**Supervision:** Changsoo Kim.

**Writing – original draft:** Wijeong Jang, Myungsok Oh.

**Writing – review & editing:** Changsoo Kim.

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
