## [Decision Letter · Decision Letter 0]

24 Nov 2022

PONE-D-22-29175Drosophila Pain Sensitization and Modulation Unveiled by a Novel Pain Model and Analgesic DrugsPLOS ONE

Dear Dr. Kim,

Thank you for submitting your manuscript to PLOS ONE. After careful consideration, we feel that it has merit but does not fully meet PLOS ONE’s publication criteria as it currently stands. Therefore, we invite you to submit a revised version of the manuscript that addresses the points raised during the review process.

We look forward to receiving your revised manuscript.

Kind regards,

Sekyu Choi

Academic Editor

PLOS ONE

Journal Requirements:

"This work was supported by grants from NRF2021R1A2C1010334, NRF-2020R1I1A1A01074292 and from the National University Development Project."

"National Research Foundation (NRF) of Korea (www.nrf.re.kr) to CK (2021R1A2C1010334) and WJ (2020R1I1A1A01074292) and National University Development Project to CK."

Reviewers' comments:

Reviewer's Responses to Questions

**Comments to the Author**

1. Is the manuscript technically sound, and do the data support the conclusions?

Reviewer #1: Yes

Reviewer #2: Partly

2. Has the statistical analysis been performed appropriately and rigorously? 

Reviewer #1: Yes

Reviewer #2: Yes

3. Have the authors made all data underlying the findings in their manuscript fully available?

Reviewer #1: Yes

Reviewer #2: Yes

4. Is the manuscript presented in an intelligible fashion and written in standard English?

Reviewer #1: Yes

Reviewer #2: Yes

5. Review Comments to the Author

Reviewer #1: In this manuscript, authors develop drosophila pain model by expressing TRPV1 channel in nociceptive sensory neurons. This transgenic fly shows aversive feeding behavior on capsaicin containing food due to pain sensation. This model employs viability assay, which produce clear-cut outcome upon alterations of pain processing. As a results, authors clearly showed that pain processing of drosophila is sensitive to several categories of drugs modulating mammalian pain processing. This manuscript brings out the important notion that drosophila pain processing is comparable to those of mammals. Furthermore, I agree with the authors that the developed drosophila pain model will be valuable platform to screen pain modulating drugs. Overall, the experiments are well designed and performed. But I saw necessary information are missing in some places and I have a few concerns to be addressed before publication.

Major

1. Fig. S1A and B, describe how % response was measured. In addition, it is very hard to see what FigS3 is. Please upgrade the image by adding other part’s label, scale bar etc.

2. Fig. 2, TRPV1 is activated by noxious high temperature like above 40oC. Thus the temperature effect on hyperalgesia might be simply due to increased expression of TRPV1, given the copy number of TRPV1 affects viability of md-gal4, TRPV1 flies as shown in Fig.3. Thus, I am wondering whether the expression level of TRPV1 is not changed with the temperature.

3. Legends for supplementary information are missing.

Minor

1. Fig. S4. I don’t understand the meaning of testing berberine and pancreatine.

2. Fig4. symbols are hard to differentiate each other. Please change.

3. Fig. S3B. tick and tick label are not aligned. Fig. S3. Why no data for diclofenac??

4. Text for Fig. 7B. Although authors describe the experimental condition in the figure legend, it is a bit confusing whether morphine was treated following pretreatment or not in the text without the full description of the experiments.

Reviewer #2: Summary: This study from the Kim group develops a novel gain of function assay (overexpression of human TRPV1 and exposure to capsaicin) that may have some utility in study of nociception in Drosophila. This is primary research that has not, to my knowledge, been reported elsewhere. As a proof of principle of the system they use the system to document effects of known pharmacological analgesics that work in humans. Some essential controls are missing from some of the experiments and the use of adult survival as an easy-to-read proxy for nociceptive effects has potential interpretation issues that are worth resolving. There are other conceptual/genetic issues elaborated below. An argument could be made that this would be a stronger paper if focused on the genes/drugs where the presumed target is more clearly known and all the controls are performed. If these issues can be addressed this paper will make a nice addition to the literature of nociception in the fly.

Major Points:

1. Essential controls are missing from Gal4/UAS experiments. Example: All of the experiments in Figure 1 are using the md-Gal4> UAS-TRPV1 X 3 genotype. Interpretation depends upon the Gal4 alone and UAS-TRPV1 X 3 alone genotypes not showing decreased feeding bouts (B) and decreased % eating (C) on capsaicin-containing food. Is this the case? These controls are essential to show that the transgene insertion sites themselves are not affecting the biology being observed.

2. The various dose responses in Figs 2 and 3 (capsaicin concentration dependence, temp dependence, TRPV1 dosage dependence) are reassuring. That said, it would be vastly more reassuring if the readout was directly a nociceptive one since we cannot be certain of the reason(s) for decreased survival on capsaicin when expressing TRPV1. Quantitation of the amplitude or duration of feeding bouts or the exaggerated facial rubbing/grooming behavior seen in the movies would be more convincing. Survival as a proxy for nociceptive effects is more problematic with the pharmacologic experiments. Without a dose-response curve of adult survival on simple exposure to varying amounts of these analgesics (not shown) it is difficult to interpret these data. Although the authors are looking at a rescue of an acute survival defect (reassuring since survival is being enhanced) it would be helpful to know that the effective doses of the drugs (those where increased survival is being observed) do not themselves have an effect on survival or feeding behavrior in controls (Gal4 alone, UAS-alone, or Canton S flies).

3. The ascending/descending pathway claims are greatly overstated. The evidence for descending pathways in flies is scant as yet (only the Neely paper ref #45). And the action of these drugs on the ascending pathway (where relevant) is assumed from other biology. Section titles saying “analgesics that inhibit the ascending/descending pain pathway” should at least be toned down “analgesics presumed to inhibit…” or “that might inhibit…” A presumed drug action in vertebrates cannot be simply transposed to the fly system. Example from the literature: Substance P, which is a pain neurotransmitter from primary sensory neurons in vertebrates, is not expressed in nociceptors in fly larvae (Im et al, eLIfe, 2015). In flies the Tachykin (Substance-P-like peptide) comes from elsewhere, neurosecretory cells in the CNS, and acts on its receptor in the sensory neurons. There could easily be other differences in mechanism of action that are similar to this. Note: morphine does not act only on the descending pathway. One of the reasons it is a really good analgesic is that it also has dampening effects on primary nociceptor output, at least in vertebrates.

4. Some discussion of the putative fly genetic targets is merited, especially for the drugs where this is not obvious. The only drug for which the fly target is presumably clear is gabapentin/straightjacket. And even here in the fly system no group has shown actual binding of gabapentin to the fly receptor. Figure 4 B is nice but there is no gabapentin in this experiment so we can’t tell if the gabapentin-prolonged survival effect in 4A depends on straightjacket levels. Some discussion of the presumed pharmacologic/genetic targets of the other drugs and their likely orthologs in flies would be helpful in the absence of experiments showing that these drugs depend on these target gene levels for their effects. This is particularly needed for morphine, because the field has long accepted, based on bioinformatic comparison of GPCR sequences, that flies do not have obvious orthologs of vertebrate mu/kappa/delta opiate receptors.

5. The morphine effect shown is not tolerance. Tolerance would be decreased analgesia on repeated/repetitive administration of the same dose of morphine. That is not the format of this experiment, which is more akin to a simple dose-response.

6. Do any of the drugs inhibit or enhance feeding behavior on normal food? If so this might complicate interpretation.

Minor comments:

1. Figure 3B, logically, belongs in Figure 2. It is another way of demonstrating dose-response.

2. Does the inhibition of feeding bouts on capsaicin continue past 250 s? This is pretty important for the starvation/lethality argument… One might presume that the flies would try to feed again as they become hungrier and hungrier with time.

3. Is the rapid decrease in viability seen in Figure 1 faster than that that would be seen if just simply starving the flies? Like control flies with no food at all?

4. Methods: The selection marker and verification of transgene presence should be included in the “Genetics and fly strains” section describing construction of UAS-TRPV1. It is not enough to say “were recovered by microinjection”.

5. The need for 3 copies of the UAS-TRPV1 transgene for full effects hinders the utility of the system to some extent, as it would require having three copies present in any future scheme involving genetic analysis/screening.

6. Just curious: Will adult flies of md-Gal4 > UAS-TRPV1 x 3 go back to eating food if returned to normal food before they starve? Is there a time cutoff of exposure to capsaicin food beyond which they cannot recover?

7. Results, second paragraph: “on the middle” is not a scientific anatomical term. Which segment range, specifically?

6. PLOS authors have the option to publish the peer review history of their article (what does this mean?). If published, this will include your full peer review and any attached files.

Reviewer #1: No

Reviewer #2: No

---

## [Author Response · Author response to Decision Letter 0]

16 Jan 2023

Reviewer #1: In this manuscript, authors develop drosophila pain model by expressing TRPV1 channel in nociceptive sensory neurons. This transgenic fly shows aversive feeding behavior on capsaicin containing food due to pain sensation. This model employs viability assay, which produce clear-cut outcome upon alterations of pain processing. As a results, authors clearly showed that pain processing of drosophila is sensitive to several categories of drugs modulating mammalian pain processing. This manuscript brings out the important notion that drosophila pain processing is comparable to those of mammals. Furthermore, I agree with the authors that the developed drosophila pain model will be valuable platform to screen pain modulating drugs. Overall, the experiments are well designed and performed. But I saw necessary information are missing in some places and I have a few concerns to be addressed before publication.

Major

1. Fig. S1A and B, describe how % response was measured. In addition, it is very hard to see what FigS3 is. Please upgrade the image by adding other part’s label, scale bar etc.

---corrected as suggested. 

2. Fig. 2, TRPV1 is activated by noxious high temperature like above 40oC. Thus the temperature effect on hyperalgesia might be simply due to increased expression of TRPV1, given the copy number of TRPV1 affects viability of md-gal4, TRPV1 flies as shown in Fig.3. Thus, I am wondering whether the expression level of TRPV1 is not changed with the temperature.

--- We removed the sentence “viability on capsaicin food is determined by factors (capsaicin and heat) that stimulate TRPV1”.

--- Our RT-PCR showed that TRPV1 transcripts were not increased at 29oC compared to 25oC. In the paper (Seroude et al., 2002) , 3/4 Gal4/UAS-lacZ have higher expression of betagalatosidase at 29oC compared to 25oC, but 1/4 Gal4/UAS-lacZ line did not exibit higher beta-galactosidase at 29oC. So it is not that all Gal4/UAS-gene express more UAS-gene at 29oC compared to 25oC. 

---We observed dramatic effect of temperature on the viability of md-TRPV1(3) on capsaicin food. This is because, molecules, in particular, small molecules (neurotransmitter), have a higher thermal energy at higher temperature. So neural excitation, synaptic transmission involving md-Gal4>UAS-TRPV1 system will be faster and stronger at high temperature. This temperature effect on physiology is shown in other case. For example, flies exhibit lifespan of ~ 60 days at 25 oC while lifespan of ~ 30 days at 29oC. 

3. Legends for supplementary information are missing.

Minor

1. Fig. S4. I don’t understand the meaning of testing berberine and pancreatine.

--- These figures are removed.

2. Fig4. symbols are hard to differentiate each other. Please change.

---corrected.

3. Fig. S3B. tick and tick label are not aligned. Fig. S3. Why no data for diclofenac??

---corrected.

--- diclofenac is included in Fig S7.

4. Text for Fig. 7B. Although authors describe the experimental condition in the figure legend, it is a bit confusing whether morphine was treated following pretreatment or not in the text without the full description of the experiments.

--- corrected. Yes, morphine was pretreated (or prefed) for six days, which was included in the text.

Reviewer #2: Summary: This study from the Kim group develops a novel gain of function assay (overexpression of human TRPV1 and exposure to capsaicin) that may have some utility in study of nociception in Drosophila. This is primary research that has not, to my knowledge, been reported elsewhere. As a proof of principle of the system they use the system to document effects of known pharmacological analgesics that work in humans. Some essential controls are missing from some of the experiments and the use of adult survival as an easy-to-read proxy for nociceptive effects has potential interpretation issues that are worth resolving. There are other conceptual/genetic issues elaborated below. An argument could be made that this would be a stronger paper if focused on the genes/drugs where the presumed target is more clearly known and all the controls are performed. If these issues can be addressed this paper will make a nice addition to the literature of nociception in the fly.

Major Points:

1. Essential controls are missing from Gal4/UAS experiments. Example: All of the experiments in Figure 1 are using the md-Gal4> UAS-TRPV1 X 3 genotype. Interpretation depends upon the Gal4 alone and UAS-TRPV1 X 3 alone genotypes not showing decreased feeding bouts (B) and decreased % eating (C) on capsaicin-containing food. Is this the case? These controls are essential to show that the transgene insertion sites themselves are not affecting the biology being observed.

---control data are included in figS2.

2. The various dose responses in Figs 2 and 3 (capsaicin concentration dependence, temp dependence, TRPV1 dosage dependence) are reassuring. That said, it would be vastly more reassuring if the readout was directly a nociceptive one since we cannot be certain of the reason(s) for decreased survival on capsaicin when expressing TRPV1. Quantitation of the amplitude or duration of feeding bouts or the exaggerated facial rubbing/grooming behavior seen in the movies would be more convincing. Survival as a proxy for nociceptive effects is more problematic with the pharmacologic experiments. Without a dose-response curve of adult survival on simple exposure to varying amounts of these analgesics (not shown) it is difficult to interpret these data. Although the authors are looking at a rescue of an acute survival defect (reassuring since survival is being enhanced) it would be helpful to know that the effective doses of the drugs (those where increased survival is being observed) do not themselves have an effect on survival or feeding behavrior in controls (Gal4 alone, UAS-alone, or Canton S flies).

--- The data are included in figS8.

Flies eat well and live well on food supplemented with these drugs. 

3. The ascending/descending pathway claims are greatly overstated. The evidence for descending pathways in flies is scant as yet (only the Neely paper ref #45). And the action of these drugs on the ascending pathway (where relevant) is assumed from other biology. Section titles saying “analgesics that inhibit the ascending/descending pain pathway” should at least be toned down “analgesics presumed to inhibit…” or “that might inhibit…” A presumed drug action in vertebrates cannot be simply transposed to the fly system. Example from the literature: Substance P, which is a pain neurotransmitter from primary sensory neurons in vertebrates, is not expressed in nociceptors in fly larvae (Im et al, eLIfe, 2015). In flies the Tachykin (Substance-P-like peptide) comes from elsewhere, neurosecretory cells in the CNS, and acts on its receptor in the sensory neurons. There could easily be other differences in mechanism of action that are similar to this. Note: morphine does not act only on the descending pathway. One of the reasons it is a really good analgesic is that it also has dampening effects on primary nociceptor output, at least in vertebrates.

--- corrected as suggested.

4. Some discussion of the putative fly genetic targets is merited, especially for the drugs where this is not obvious. The only drug for which the fly target is presumably clear is gabapentin/straightjacket. And even here in the fly system no group has shown actual binding of gabapentin to the fly receptor. Figure 4 B is nice but there is no gabapentin in this experiment so we can’t tell if the gabapentin-prolonged survival effect in 4A depends on straightjacket levels. Some discussion of the presumed pharmacologic/genetic targets of the other drugs and their likely orthologs in flies would be helpful in the absence of experiments showing that these drugs depend on these target gene levels for their effects. This is particularly needed for morphine, because the field has long accepted, based on bioinformatic comparison of GPCR sequences, that flies do not have obvious orthologs of vertebrate mu/kappa/delta opiate receptors.

---The Drosophila presumed targets of drugs are included in discussion.

5. The morphine effect shown is not tolerance. Tolerance would be decreased analgesia on repeated/repetitive administration of the same dose of morphine. That is not the format of this experiment, which is more akin to a simple dose-response.

--- The word “morphine tolerance” is removed. 

6. Do any of the drugs inhibit or enhance feeding behavior on normal food? If so this might complicate interpretation.

--- Fig S8 is included, showing no effect on viability on food supplemented with these drugs. These drugs at the concentration effective in reducing pain do not affect feeding.

Minor comments:

1. Figure 3B, logically, belongs in Figure 2. It is another way of demonstrating dose-response.

--- Corrected as suggested, namely, Figure 3B is combined to Figure 2.

2. Does the inhibition of feeding bouts on capsaicin continue past 250 s? This is pretty important for the starvation/lethality argument… One might presume that the flies would try to feed again as they become hungrier and hungrier with time.

--- We watched over 1.5 hrs. After a few sipping on capsaicin food, they did not attempt to try to sip in this period (1.5 hour). 

We removed the sentence “eventually stopped returning to the food” from results and “ with the flies eventually stopping all sipping“ from discussion, because we did not watch 24 hours or more. 

3. Is the rapid decrease in viability seen in Figure 1 faster than that that would be seen if just simply starving the flies? Like control flies with no food at all?

---The rapid decrease in viability is similar to starved flies, which is shown in Fig S5A. 

4. Methods: The selection marker and verification of transgene presence should be included in the “Genetics and fly strains” section describing construction of UAS-TRPV1. It is not enough to say “were recovered by microinjection”.

---corrected. 

5. The need for 3 copies of the UAS-TRPV1 transgene for full effects hinders the utility of the system to some extent, as it would require having three copies present in any future scheme involving genetic analysis/screening.

--- md-TRPV1(3) (genotype--- w; md-Gal4, UAS-TRPV1/CyO; UAS-TRPV1/UAS-TRPV1) is maintained as a stable line and used for drug evaluation. 

--- For UAS-RNAi screening to identify pain genes, md-TRPV1(3) flies are crossed to UAS-RNAi (experimental) or w1118 (control), to produce md-TRPV1(2) with or without UAS-RNAi. And then score viability. It is easy and simple to identify pain genes.

w; md-Gal4, UAS-TRPV1/+; UAS-TRPV1/+ (control two copies)

w; md-Gal4, UAS-TRPV1/+; UAS-TRPV1/UAS-stj RNAi. (experimental with UAS-RNAi)

6. Just curious: Will adult flies of md-Gal4 > UAS-TRPV1 x 3 go back to eating food if returned to normal food before they starve? Is there a time cutoff of exposure to capsaicin food beyond which they cannot recover?

---Fig S6 shows the data. Yes, they eat well and live well on food lacking capsaicin

7. Results, second paragraph: “on the middle” is not a scientific anatomical term. Which segment range, specifically?

--- corrected. abdominal segment five.

---

## [Decision Letter · Decision Letter 1]

2 Feb 2023

Drosophila Pain Sensitization and Modulation Unveiled by a Novel Pain Model and Analgesic Drugs

PONE-D-22-29175R1

Dear Dr. Kim,

We’re pleased to inform you that your manuscript has been judged scientifically suitable for publication and will be formally accepted for publication once it meets all outstanding technical requirements.

Kind regards,

Sekyu Choi

Academic Editor

PLOS ONE

Additional Editor Comments (optional):

Reviewers' comments:

Reviewer's Responses to Questions

**Comments to the Author**

1. If the authors have adequately addressed your comments raised in a previous round of review and you feel that this manuscript is now acceptable for publication, you may indicate that here to bypass the “Comments to the Author” section, enter your conflict of interest statement in the “Confidential to Editor” section, and submit your "Accept" recommendation.

Reviewer #1: All comments have been addressed

Reviewer #2: All comments have been addressed

2. Is the manuscript technically sound, and do the data support the conclusions?

Reviewer #1: Yes

Reviewer #2: Yes

3. Has the statistical analysis been performed appropriately and rigorously? 

Reviewer #1: Yes

Reviewer #2: Yes

4. Have the authors made all data underlying the findings in their manuscript fully available?

Reviewer #1: Yes

Reviewer #2: Yes

5. Is the manuscript presented in an intelligible fashion and written in standard English?

Reviewer #1: Yes

Reviewer #2: Yes

6. Review Comments to the Author

Reviewer #1: The authors have addressed my concerns and comments with textual edits in their revised manuscript. I recommend this manuscript be published in Plos One.

Reviewer #2: The revisions to the paper satisfactorily address the various issues with control/context experiments and terminology. The section in the discussion on possibile targets is a nice addition.

7. PLOS authors have the option to publish the peer review history of their article (what does this mean?). If published, this will include your full peer review and any attached files.

Reviewer #1: No

Reviewer #2: No

---

## [Editor Report · Acceptance letter]

7 Feb 2023

PONE-D-22-29175R1 

*Drosophila* Pain Sensitization and Modulation Unveiled by a Novel Pain Model and Analgesic Drugs 

Dear Dr. Kim:

I'm pleased to inform you that your manuscript has been deemed suitable for publication in PLOS ONE. Congratulations! Your manuscript is now with our production department. 

Kind regards, 

on behalf of

Dr. Sekyu Choi 

Academic Editor

PLOS ONE